# A Bayesian Fast-Slow Framework to Mitigate Interference in Non-Stationary Reinforcement Learning

**Yihuan Mao**
Institute for Interdisciplinary Information Sciences
Tsinghua University
maoyh1024@gmail.com

**Chongjie Zhang**
Department of Computer Science & Engineering
Washington University in St. Louis
chongjie@wustl.edu

## Abstract

Given the ever-changing nature of the world and its inhabitants, agents must possess the ability to adapt and evolve over time. Recent research in Given the ever-changing nature of the world and its inhabitants, agents must possess the ability to adapt and evolve over time. Recent research in non-stationary MDPs has focused on addressing this challenge, providing algorithms inspired by task inference techniques. However, these methods ignore the detrimental effects of interference, which particularly harm performance in contradictory tasks, leading to low efficiency in some environments. To address this issue, we propose a Bayesian Fast-Slow Framework (BFSF) that tackles both cross-task generalization and resistance to cross-task interference. Our framework consists of two components: a 'fast' policy, learned from recent data, and a 'slow' policy, learned through meta-reinforcement learning (meta-RL) using data from all previous tasks. A Bayesian estimation mechanism determines the current choice of 'fast' or 'slow' policy, balancing exploration and exploitation. Additionally, in the 'fast' policy, we introduce a dual-reset mechanism and a data relabeling technique to further accelerate convergence when encountering new tasks. Experiments demonstrate that our algorithm effectively mitigates interference and outperforms baseline approaches. Code is available at https://github.com/cedesu/BFSF.

Reinforcement Learning (RL) in non-stationary environments has long attracted significant attention, leading to the emergence of research areas such as continual RL [30, 19] and non-stationary MDPs [7, 24, 31]. These areas approach challenges from different perspectives. For example, catastrophic forgetting, a well-known issue in continual RL, arises due to limited memory storage. In contrast, research on non-stationary MDPs focuses on understanding the underlying dynamics of the environment to facilitate better adaptation across varying contexts.

One critical challenge in non-stationary environments is interference, where the learning process is negatively impacted by experiences from previous tasks, as illustrated in Figure 1. This interference arises primarily because task boundaries are either unknown or absent in the streaming task setting. Many real-world problems exhibit such non-stationarity and suffer from interference. For instance, a UAV must adapt its behavior under varying weather conditions [27, 26]. Similarly, the evolving regime of the stock market can be viewed as a time-varying environment, where an effective strategy

39th Conference on Neural Information Processing Systems (NeurIPS 2025).

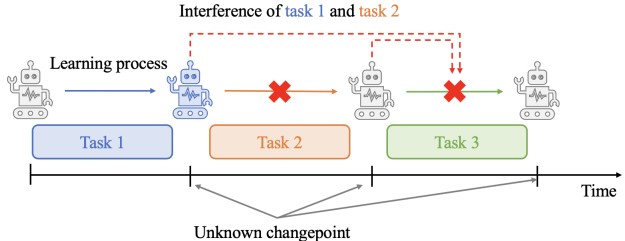

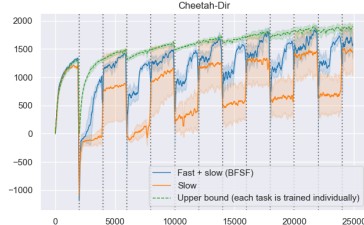

(a) The diagram of interference in non-stationary MDPs.

(b) The learning curve of how BFSF alleviates the problem of interference.

Figure 1: Figure 1(a) illustrates the interference problem in non-stationary MDPs. The agent learns to perform well on the current task, but the changepoint is unknown. As a result, when the agent begins learning a new task (e.g., task 2), experience from previous tasks (e.g., task 1) can hinder performance. This interference phenomenon also occurs across consecutive tasks. To address this, we propose BFSF, which incorporates a 'fast' policy that learns from recent data to mitigate interference, alongside a 'slow' policy using meta-RL to learn a context-based policy from all previous tasks. Figure 1(b) demonstrates BFSF's ability to resist interference in the Cheetah-Dir task, which involves two contradictory tasks: moving forward and backward. In the second phase, the learning curve shows less disruption compared to the 'slow' policy only. The highest performance throughout the non-stationary MDP process is close to the upper bound, which represents the scenario where the two tasks are trained separately.

must stay adaptive while leveraging past experience [13, 3]. These real-world scenarios underscore the urgent need for a framework that can operate efficiently in non-stationary MDPs with interference.

Despite its significant negative impact on performance, the issue of interference has been largely overlooked in the literature. Some existing works [18] address interference from the perspective of representation, while others [20] discuss the inverse interference of current tasks on previously learned tasks, a phenomenon referred to as catastrophic forgetting in continual RL. This lack of attention to cross-task interference is concerning, as it can severely degrade performance in successive tasks. In this work, we specifically analyze the effects of interference and propose effective strategies to mitigate its detrimental impacts.

To address the interference problem, we introduce the Bayesian Fast-Slow Framework (BFSF). This framework dynamically selects between two learning strategies: a 'fast' policy, which quickly adapts to new tasks using recent data, and a 'slow' policy, which is learned through meta-RL and captures knowledge from historical data. Unlike previous approaches that focus solely on latter, often leading to severe interference in the face of sudden task changes, our framework not only mitigates interference but also preserves the advantages of meta-RL. A Bayesian estimation mechanism is employed in each epoch to decide which policy, fast or slow, is more promising based on recent history. Only the most recent returns are used to update the Bayesian estimates, ensuring that outdated data does not influence the decision.

We also identify that the 'fast' policy can sometimes underperform. One reason is that neural networks often experience performance degradation when trained on data from different distributions, a common issue in non-stationary environments. To address this, we introduce a dual-reset mechanism that periodically reinitializes one of the dual networks to prevent degradation, while alternating between the networks to ensure stable performance. Another challenge is that learning from scratch typically requires extensive online interaction. To mitigate this, we propose data relabeling, utilizing historical data from previous tasks to enhance learning efficiency and improve performance in few-shot settings when facing new tasks.

In summary, our contributions are twofold: i) We analyze the interference problem in non-stationary MDPs. ii) We propose the Bayesian Fast-Slow Framework (BFSF), which combines a fast policy, enhanced by a dual-reset mechanism and data relabeling, to efficiently handle recent tasks, and a slow policy for cross-task generalization. Through Bayesian estimation, we effectively address interference and improve overall performance. Experimental results demonstrate BFSF's superiority in resisting interference and outperforming baseline methods across various non-stationary environments.

# 1 Preliminaries

**Notations and problem definition**  A Markov Decision Process (MDP) is defined as $M = \langle S, A, P, R \rangle$, where $S$ and $A$ represent the state and action spaces, respectively. The transition function of the environment is denoted as $P$, and $R$ represents the reward function. The expected return of a policy is given by $\mathbb{E}[\sum_{t=0}^{\infty} R_t]$. In non-stationary MDPs, the underlying MDP evolves over time. These changes can occur sequentially, such as $M_1, M_2, \cdots$, or gradually over time. The objective is to maximize the expected return, $\mathbb{E}[\sum_{t=0}^{\infty} R_t]$ with the evolving dynamics of the MDP.

**Context-based policy**  In a standard MDP, the policy function is defined as $\pi(a|s)$, which determines the probability of selecting action $a$ given state $s$. In non-stationary settings, a context-based policy is introduced to adapt to varying environments. This policy is denoted as $\pi(a|s, c)$, where $c$ is the context, a set of trajectories related to the current environment. A trajectory consists of the sequence of states, actions, and rewards at each timestep, expressed as $s_0, a_0, r_0, s_1, a_1, r_1, s_2, a_2, r_2, \ldots$. In implementation, the context is collected from recent interactions, reflecting the underlying MDP.

The technique of learning the context-based policy has been extensively studied in the field of meta-reinforcement learning (meta-RL) [24, 37]. However, meta-RL differs from the non-stationary MDPs setting in this work. In meta-RL, task information is explicitly available, and there is no continuous adaptation process. Context-based meta-RL methods typically map the contextual information, often represented as transition data, into a latent space $\mathcal{Z}$. By assigning a latent variable $z \in \mathcal{Z}$ to represent the task, this approach effectively frames the problem as a partially observable MDP (POMDP) [16], where $z$ constitutes the unobserved portion of the state. In meta RL, PEARL learns the posterior distribution $q(z|c)$, which means the posterior latent variable distribution given the context $c$, and uses posterior sampling to sample $z$ to integrate these latent variables with off-policy RL algorithms.

---

**Algorithm 1** Bayesian fast-slow framework (BFSF)

---

1: Input: A 'fast' policy $\pi_{fast}$ (including the dual policies $\pi_{fast}^{(1)}, \pi_{fast}^{(2)}$), a 'slow' policy $\pi_{slow}$, the number of epochs $E$, the window of recent data $w$
2: Initialize return list $\{R_i\}_{i \in 1 \cdots E}$ and choice list $\{\text{choice}_j\}_{i \in 1 \cdots E}$
3: **for** epoch $e = 1 \cdots E$ **do**
4:     # Bayesian inference of the expected return
5:     $\hat{R}_{fast} = \text{Posterior}(\{R_{i \in [e-w,e]} | \text{choice}_i = fast\})$
6:     $\hat{R}_{slow} = \text{Posterior}(\{R_{i \in [e-w,e]} | \text{choice}_i = slow\})$
7:     # Online interaction
8:     **if** $\hat{R}_{fast} > \hat{R}_{slow}$ **then**
9:         $\text{choice}_e := fast$
10:         Collect data using $\pi_{fast}$
11:     **else**
12:         $\text{choice}_e := slow$
13:         Collect data using $\pi_{slow}$
14:     **end if**
15:     # Training
16:     Update $\pi_{fast}$ by Algorithm 2
17:     Update $\pi_{slow}$ by the meta-RL algorithm
18: **end for**

---

# 2 Bayesian Fast-Slow Framework (BFSF)

The Bayesian Fast-Slow Framework (BFSF) is designed to mitigate interference by dynamically deploying either a 'fast' policy, which learns from recent data, or a 'slow' policy, trained using meta-RL principles. The term 'fast' arises from its 'fast-adaptation' ability to learn directly and efficiently from recent data. In contrast, the 'slow' policy enables cross-task understanding and generalization, which may hinder training speed, especially when the number of observed tasks is limited in early phase. The decision is made based on Bayesian estimation of current expected return.

As illustrated in Algorithm 1, during each epoch, Bayesian inference is applied to estimate the posterior expected return using the recent return history, for both the fast and slow policies. Let $R_i$ denote the return obtained in the $i$-th epoch, and $\text{choice}_i$ indicate whether the 'fast' or 'slow' policy was selected during that epoch. During the online interaction phase, the policy with the higher estimated posterior value is selected, aiming to generate higher-quality experience. At the end of each epoch, both the fast and slow policies are updated using their respective replay buffers. The detailed computation of the Bayesian posterior, described in lines 5 and 6 of Algorithm 1, is elaborated in Section 2.1. In addition to the online interaction and Bayesian estimation, the training process for the 'fast' policy is detailed in Algorithm 2, while the 'slow' policy is trained according to the context-based meta-RL algorithm PEARL [24], which is one of the first context-based methods and serves as the baseline for numerous subsequent works. The visualization of choosing the 'fast' or 'slow' policy is provided in Appendix E.

## 2.1 Bayesian Inference

The detailed update rule for Bayesian posterior estimation is outlined below. For simplicity, assume that the recent returns of a given policy, $R_{i_1}, R_{i_2}, \ldots$, are approximately drawn from a normal distribution $\mathcal{N}(\mu, 1/\phi)$, where $\phi$ is a constant. While this assumption is commonly used, other distributional forms could also be considered depending on the context. The prior distribution for the parameter $\mu$ is assumed to follow $\mu \sim \mathcal{N}(\mu_0, 1/\phi_0)$. The posterior estimation of $\mu$ then follows the standard derivation below, as detailed in Appendix D.

$$
\begin{aligned}
p(\mu | \{R_{i_1}, R_{i_2}, \cdots\}, \mu_0, \phi_0) &\sim \mathcal{N}(\mu_1, 1/\sigma_1^2), \\
\text{where } \mu_1 = \frac{\phi_0 \mu_0 + n\phi \overline{R}}{\phi_0 + n\phi}&, \sigma_1^2 = \frac{1}{\phi_0 + n\phi}.
\end{aligned}
\tag{1}
$$

To better interpret the result of Bayesian inference, note that the posterior mean $\mu_1$ can be decomposed:

$$
\mu_1 = \frac{\phi_0 \mu_0 + n\phi \overline{R}}{\phi_0 + n\phi} = \frac{\phi_0}{\phi_0 + n\phi} \mu_0 + \frac{n\phi}{\phi_0 + n\phi} \overline{R}.
\tag{2}
$$

It shows that the posterior mean $\mu_1$ is a weighted average of the prior mean $\mu_0$ and the sample average $\overline{R}$. As more samples are collected, the weight shifts toward trusting the sample average $\overline{R}$. Conversely, when only a few samples are available, the posterior relies more heavily on the prior $\mu_0$.

## 2.2 'Fast' Policy Learning

The 'fast' policy, learned from recent data, is critical for mitigating interference in non-stationary MDPs. However, the standard learning paradigm often encounters challenges under these conditions. To address these issues, we propose specific structural designs that significantly enhance the efficiency and adaptability of the 'fast' policy.

---

**Algorithm 2** Training process of the 'fast' policy.

---

1: Input: The 'fast' policy $\pi_{fast}$ (including the dual policies $\pi_{fast}^{(1)}, \pi_{fast}^{(2)}$), a contextual dynamics model $M$, current epoch $e$, reset frequency $\nu$
2: Output: The updated $\pi_{fast}$
3: # Dual-reset mechanism
4: **if** $e \bmod \nu = 0$ **then**
5:    $\pi_{fast}^{(1)}, \pi_{fast}^{(2)} = \pi_{fast}^{(2)}, \text{Init}(\pi_{fast}^{(1)})$
6: **end if**
7: # Data relabeling
8: Relabel the recent data by $M$ using the recent trajectories as the context.
9: # Training process
10: Train $\pi_{fast}^{(1)}, \pi_{fast}^{(2)}$ using the relabeled data

---

**Dual-reset Mechanism**    A key challenge in continual learning is the performance degradation of neural networks when trained on successive tasks. One common observation is that the learning curve for the second task often struggles to converge to an optimal point, even when task difficulty is comparable (as detailed in Appendix C.2). This phenomenon is also noted and studied in ITER [14].

To address this, we propose the dual-reset mechanism, as outlined in Algorithm 2, which mitigates performance degradation by periodically reinitializing the model. However, to avoid the inferior performance typically observed immediately after reinitialization, we introduce a dual-model system. This ensures that during any interaction phase, even directly after initialization, a fully trained model is always available for deployment.

**Data Relabeling**    Another challenge lies in the limited recent data available for the 'fast' policy, which is necessary for rapid adaptation in non-stationary environments. This small dataset size may not support learning a robust policy, especially over prolonged training periods, in contrast to the 'slow' policy that can utilize the entire historical dataset. As a result, the 'fast' policy tends to perform significantly worse, as shown in Figure 5. To address this limitation, we incorporate data relabeling, which significantly enhances the amount of usable data, enabling the learning of a stronger policy.

Specifically, the data relabeling process relies on maintaining a context-based dynamics model. This model takes the context, state, and action as input, and outputs the relabeled next state and reward. Leveraging the context-based property, it becomes possible to relabel historical data from other tasks into the context of the current task. By combining relabeled data with the original dataset, the learning efficiency of the 'fast' policy is substantially improved. Further details can be found in Appendix C.1.

### 2.3   Theoretical Analysis

In this section, we provide a theoretical analysis of the sub-optimality bound of the Bayesian Fast-Slow Framework (BFSF) and present Theorem 2.1.

**Theorem 2.1.**

$$
\begin{aligned}
&\text{Suboptimality}(BFSF) \\
&\leq [\mathcal{D}_{\ell_1}(p_M, p_{M'})(r_{max} + V_{max}) + r_{diff}]H \\
&+ |U_{r,r'}(\pi_{M'}^*)| + \frac{1}{2}V_{max}\mathcal{D}_{\ell_1}(p_{M'}(s,a), p_M(s,a)),
\end{aligned}
\tag{3}
$$

*where $M, M'$ denote the original and relabeled MDPs, and $p_M, p_{M'}$ are their transition functions. $H$ is the horizon, $r_{max}, V_{max}$ are the maximum reward and value, $r_{diff}$ represents the maximum reward gap between $M, M'$, $U_{r,r'}(\pi)$ is defined as $\mathbb{E}_{(s,a)\sim\rho_M^\pi}[r'(s,a) - r(s,a)]$, and $\mathcal{D}_{\ell_1}$ is the L1 distance.*

The following provides a proof sketch and interpretation of Theorem 2.1. First, the suboptimality is defined as the minimum between the fast and slow policies, with the Bayesian estimation serving as an unbiased estimate. We focus on the suboptimality of the 'fast' policy, $|\eta_M(\pi_M^*) - \eta_M(\pi_{M'}^*)|$. This suboptimality can then be decomposed into several components as detailed in Appendix A.1. For ease of comprehension, the first term represents the gap in optimal expected return between the relabeled and original MDPs. The second and third terms arise from the performance difference of the same policy under different dynamics. A further analysis of the bound that incorporates optimization error is provided in Appendix A.2.

## 3   Experiments

The Bayesian Fast-Slow Framework (BFSF) is designed to address the complexities of non-stationary MDPs, specifically tackling the interference issue while ensuring generalization across experiences from different tasks. In this section, we focus on two main questions: i) How does BFSF overcome interference in non-stationary environments? ii) How does BFSF perform in real-world scenarios? iii) How do the individual components of BFSF contribute to improving performance?

To answer the first question, we provide experimental comparisons with baselines in Section 3.1, demonstrating the superiority of BFSF in mitigating interference. For the second question, we design an infinite-world simulation to better approximate real-world conditions and evaluate the performance

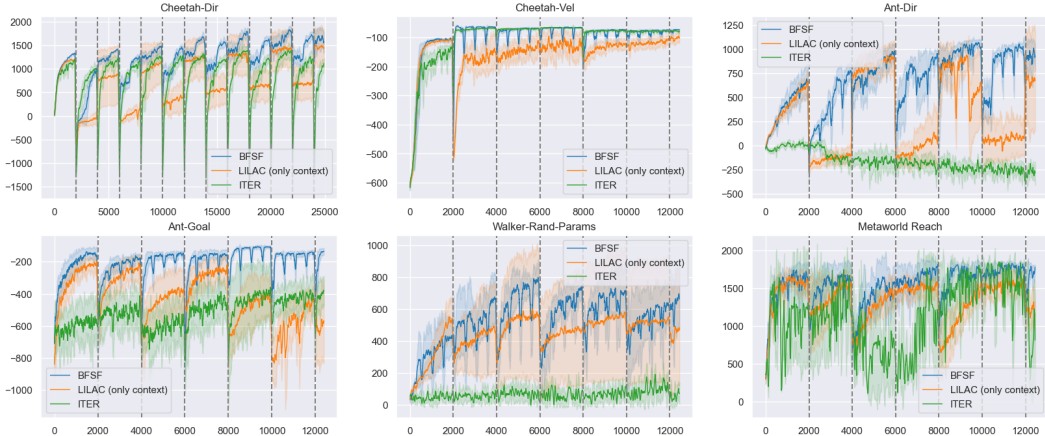

Figure 2: The learning curve of BFSF and other baselines, on the non-stationary MDPs based on 5 MuJoCo locomotion environments and 1 Meta-World environment. For clarity, we only display the curves for BFSF, LILAC, and ITER. Additional curves for CEMRL and CoMPs, along with implementation details, can be found in Appendix B.

| | CHEETAH-DIR | CHEETAH-VEL | ANT-DIR | ANT-GOAL |
|---|---|---|---|---|
| BFSF | $\mathbf{1209.0 \pm 24.0}$ | $\mathbf{-99.1 \pm 2.8}$ | $\mathbf{671.0 \pm 37.7}$ | $\mathbf{-204.9 \pm 9.3}$ |
| LILAC | $757.7 \pm 94.9$ | $-154.6 \pm 2.5$ | $318.5 \pm 66.2$ | $-426.6 \pm 4.6$ |
| ITER | $813.4 \pm 19.0$ | $-102.0 \pm 2.8$ | $-156.8 \pm 25.7$ | $-519.2 \pm 50.0$ |
| CEMRL | $852.6 \pm 36.1$ | $-196.0 \pm 5.2$ | $289.6 \pm 18$ | $-547.1 \pm 67.6$ |
| CoMPS | $277.3 \pm 40.2$ | $-117.8 \pm 6.1$ | $-142.1 \pm 6.7$ | $-602.0 \pm 72.2$ |

| | WALKER | REACH | ANT-DIR-INF | ANT-CIR-INF |
|---|---|---|---|---|
| BFSF | $530.9 \pm 59.8$ | $\mathbf{1543.4 \pm 93.5}$ | $\mathbf{153.3 \pm 4.2}$ | $\mathbf{165.2 \pm 6.8}$ |
| LILAC | $448.8 \pm 279.1$ | $1341.0 \pm 47.7$ | $85.5 \pm 18.7$ | $69.5 \pm 1.6$ |
| ITER | $65.6 \pm 3.8$ | $1112.1 \pm 168.5$ | $-338.5 \pm 64.1$ | $-337.0 \pm 69.8$ |
| CEMRL | $\mathbf{577.9 \pm 68.1}$ | $977.5 \pm 514.3$ | $-11.8 \pm 17.4$ | $28.0 \pm 13.7$ |
| CoMPS | $80.0 \pm 34.4$ | $526.9 \pm 380.0$ | $-151.0 \pm 19.4$ | $-121.1 \pm 4.8$ |

Table 1: The average return throughout the training process, comparing all the baselines. 'Walker' and 'Reach' are abbreviations for Walker-Rand-Params and Meta-World Reach, respectively.

of both BFSF and the baselines. Finally, for the third question, we conduct detailed ablation studies on the modules within BFSF, offering evidence of their effectiveness.

### 3.1 Mains Results

We evaluate the Bayesian Fast-Slow Framework (BFSF) on five MuJoCo environments and one Meta-World environment. The MuJoCo environments [28] focus on robotic locomotion and are based on the MuJoCo simulator, while Meta-World [35] is a benchmark designed for Multi-Task and meta-RL, specifically with robot manipulation tasks. These environments require adaptation across different reward functions (e.g., walking direction for Cheetah-Dir and Ant-Dir, target velocity for Cheetah-Vel, and goal location for Ant-Goal and Meta-World Reach), or across different dynamics (e.g., environment parameters for Walker-Rand-Params). These meta-RL environments are widely used in the meta-RL literature and are well-suited for non-stationary MDPs as well. We set the switching frequency of the underlying task to 2000 episodes.

We compare BFSF with four reproduced baselines. LILAC [31] is an algorithm for non-stationary MDPs that uses meta-RL techniques. It learns a latent variable to discriminate between tasks based on experiences. ITER [14] proposes an iterative approach to relearn the neural network, aiming to overcome non-stationarity. CEMRL [5] learns a task encoder from the gradients of a decoder

and provides the task encoding to downstream RL. CoMPS [4] continuously alternates between two subroutines: learning a new task using RL and performing completely offline meta-learning to prepare for subsequent task learning.

As shown in Figure 2, BFSF outperforms the baselines in non-stationary environments. The task switching is indicated by the gray dashed lines for clarity. While we conduct experiments with a fixed switching interval, it is important to note that our algorithms are designed for the general setting where the task distribution and switching timing are completely unknown to the agent. For comprehensiveness, we also experiment with an unfixed switching interval in Section 3.3.

In general, all algorithms show gradual performance improvements within a single task phase but experience a sudden performance drop immediately after task switches. This decay is expected, as the new task is unfamiliar to the agent. However, we observe that the learning curve in the second phase does not increase as quickly as in the first, which we refer to as the interference phenomenon.

BFSF effectively mitigates interference, leading to better overall performance across continual phases. LILAC, based on meta-RL methods, provides good cross-task generalization. However, it fails to address interference, resulting in slower learning during the second task. ITER's iterative relearning approach is a solid defense against interference, but relying solely on this approach leads to the learning of elementary policies, which hinders further generalization and improvement. For clarity, we only compare the curves of two baselines in the given figure, while a full comparison of average return is provided in Table 1. Full experiment results on the learning curve is provided in Appendix B.2.

## 3.2 Infinite-World Simulation

While the main results in Section 3.1 demonstrate the superiority of BFSF in mitigating interference and achieving cross-task generalization, it remains unclear how such methods perform in more realistic scenarios. In the real world, there are typically no explicit task boundaries, no fixed task initializations, and no finite set of predefined tasks. To better approximate these characteristics, we introduce an Infinite-World Simulation in MuJoCo, an environment where the agent operates in a non-episodic, continuous manner without resets.

Unlike traditional MuJoCo benchmarks, where each episode ends and resets after a fixed number of steps (e.g., every 1000 steps), our infinite-world environment allows the agent to move seamlessly through a boundless plane without ever being reinitialized. This design leads to a non-episodic interaction flow, closely mimicking the persistent nature of real-world settings. To support this, we implement a dynamic terrain loading module that handles environment generation on the fly, avoiding memory overload while preserving the illusion of an endless space.

We design two signature environments to evaluate BFSF and baselines under this setup. Ant-Dir-Inf is a non-episodic, infinite-world variant of the standard Ant-Direction environment, where the agent is required to walk alternately left and right across the plane; each time a directional goal is reached, the target direction flips. Ant-Goal-Inf is derived from Ant-Goal, where the target moves along a circular trajectory of infinite radius, requiring the agent to constantly adjust and track it over time.

The corresponding results are reported in Table 1. In addition, Figure 3 visualizes the trajectories of different methods. BFSF consistently follow the evolving goal direction, while baseline methods react more slowly. This highlights BFSF's effectiveness in non-episodic and task-free environments.

## 3.3 Ablation Studies

The ablation studies on each module of BFSF are conducted to answer the question: How do the individual components of BFSF contribute to improving performance? The results show that the 'fast' policy, as a whole, alleviates interference and enhances overall performance in non-stationary MDPs. Additionally, the dual-reset mechanism and relabeling further support the 'fast' policy by enabling more effective learning.

**Unfixed Switching Interval** We conducted experiments with an unfixed switching interval, as shown in Figure 4(a). The overall performance exhibits a pattern similar to that observed in the main experiments with a fixed switching interval: the interference issue is mitigated by BFSF, and the agent's performance continues to improve as training progresses.

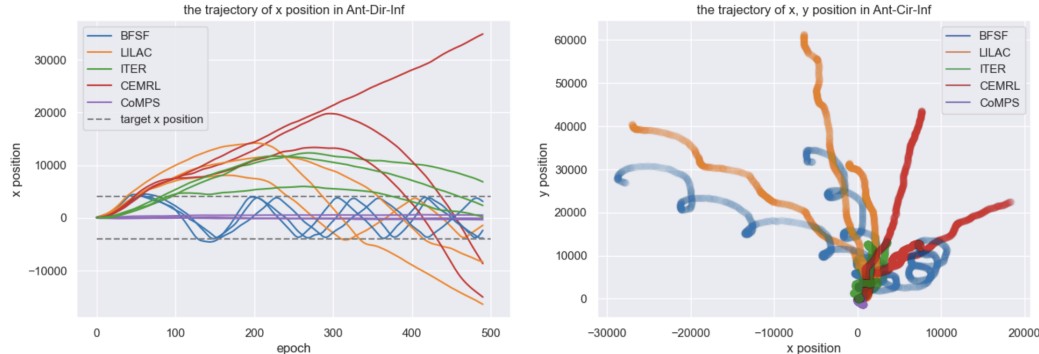

Figure 3: Visualized trajectories for Ant-Dir-Inf and Ant-Cir-Inf are shown. In Ant-Dir-Inf (left), only the BFSF algorithm successfully adapts quickly to the alternating goals between left and right directions. In Ant-Cir-Inf (right), only BFSF demonstrates rapid adaptation to the continuously moving goal along a circular path.

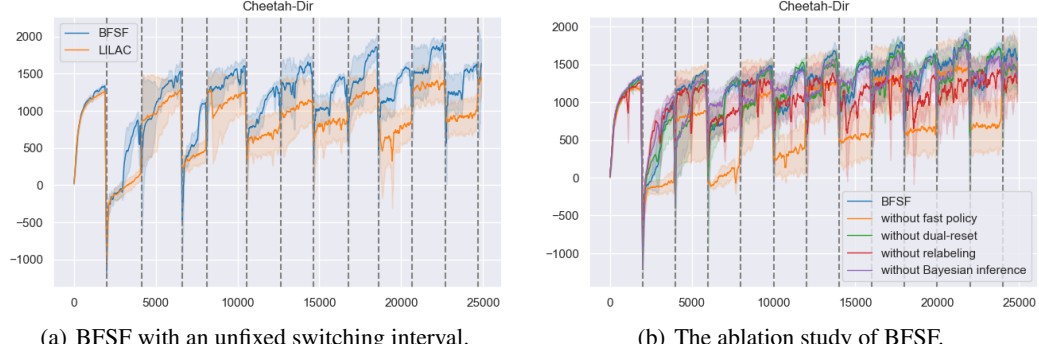

(a) BFSF with an unfixed switching interval.    (b) The ablation study of BFSF.

Figure 4: Ablation studies about the unfixed switching interval and other modules.

**'Fast' Policy Ablation**  The presence of the 'fast' policy, which learns from recent data, enables the agent to better adapt to changes in non-stationary MDPs, as shown in Figure 4(b). The curve labeled 'without fast policy' is identical to the baseline LILAC, as introduced in Section 3.1. While LILAC can generalize across tasks, it struggles to efficiently learn the optimal policy in the second task due to the interference problem. In contrast, BFSF addresses this issue, learning the second task at nearly the same speed as the first task, effectively overcoming interference. This trend is also observed in ongoing tasks, where interference does not negatively impact the performance of BFSF.

**Dual-Reset Ablation**  The dual-reset mechanism ensures the effectiveness of the 'fast' policy. As shown in Figure 4(b), without the dual-reset, the learning curve exhibits lower performance due to a suboptimal 'fast' policy.

**Relabeling Ablation**  As seen in Figure 4(b), relabeling significantly improves the learning efficiency of the 'fast' policy. With relabeling, the 'fast' policy can continuously improve its performance, even on later tasks. In contrast, BFSF without relabeling, as shown in Figure 5, struggles to achieve similar improvement in later tasks.

**Bayesian Inference Ablation**  As introduced in Section 2.1, Bayesian inference provides a suitable estimate of the expected return for both the 'fast' and 'slow' policies. Without it, the estimation becomes less effective, and proper hyperparameter tuning may be required. As illustrated in Figure 4(b), the performance and resistance to interference deteriorate in the absence of Bayesian inference.

## 4 Related works

**Non-Stationary MDPs**  Research on non-stationary MDPs primarily focuses on the challenge of recognizing potential tasks, as understanding the task transforms the non-stationary MDP into a fixed MDP. LILAC [31] first employs latent variable models to learn environment representations based on

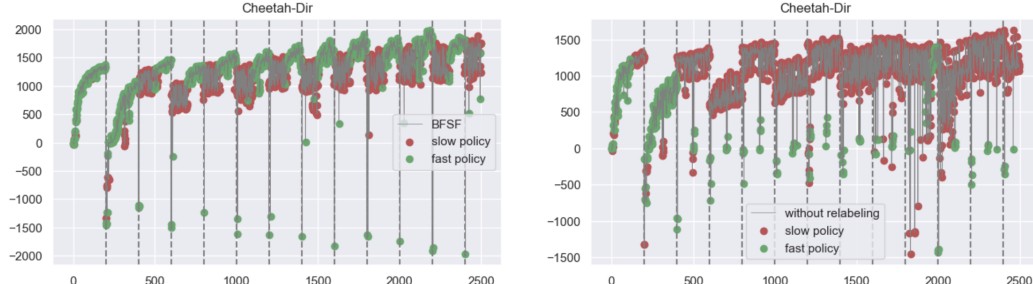

Figure 5: A comparison illustrating the performance of the 'slow' and 'fast' policies. The main difference is that with relabeling, the performance of the 'fast' policy remains higher throughout the phases, rather than significantly dropping after the initial phases.

current and past experiences, drawing inspiration from online learning and probabilistic inference. Subsequent works identified shortcomings in LILAC and proposed solutions. For instance, FANS-RL [10] models non-stationarity in terms of individual latent change factors and causal graphs. ITER [14] highlights the impact of non-stationarity on latent representations, a form of interference similar to the one discussed in our work, leading to the proposal of Iterated Relearning (ITER). Additionally, several theoretical works have also focused on non-stationary MDPs [1, 11, 2, 7]. However, none of these prior works simultaneously address both cross-task generalization and the interference problem.

Besides, continual RL shares similarities with non-stationary MDPs but focuses on different challenges, particularly catastrophic forgetting [25]. Although non-stationary MDPs and continual RL are often treated as distinct problems, their focus differs, primarily due to the assumption that task boundaries are known to agents in continual RL. As a result, research in continual RL focuses on designing submodules within the overall algorithm [30], such as replay buffers [6], network architecture [23], representation [22], or optimization strategies [21], rather than addressing the non-stationarity itself.

**Meta-RL**   Meta-RL aims to enable agents to adapt more quickly to new tasks by leveraging prior experience from multiple tasks. It bears strong resemblance to non-stationary MDPs, but assumes that agents have full access to all tasks, thus ignoring interference effects. $RL^2$ [9] learns an agent's learning algorithm, enabling it to adapt quickly to new tasks by adjusting its internal update rule based on prior experience. MAML [12], a gradient-based meta-RL algorithm, seeks a set of model parameters that can be quickly adapted to new tasks with minimal gradient updates. In contrast, context-based meta-RL methods, such as PEARL [24] and VariBAD [37], leverage contextual information to enable more efficient adaptation. While traditional meta-RL assumes access to all tasks during training, recent research has explored meta-learning in the continual task setting [4, 5], which is closely related to our work on non-stationary MDPs.

**Inference-related Works**   The issue of interference has been widely explored in related areas. In multi-task RL, task interference has been observed, and specialized network architectures have been proposed to mitigate this challenge [17, 8]. However, the source of interference in these works differs from that in non-stationary MDPs, where interference arises from unknown, streaming tasks. In continual RL, which focuses on maintaining high performance across incremental tasks, interference is also recognized and investigated at the representation level [18].

## 5   Conclusion

Non-stationarity poses a significant challenge when deploying RL agents in real-world environments. In addition to cross-task generalization through context-based algorithms, a challenge that has been thoroughly explored in previous works, we have identified that interference can severely hinder performance, especially when tasks conflict with each other. To address this issue, we introduce the Bayesian Fast-Slow Framework (BFSF), which incorporates a 'fast' policy that learns from recent history to prevent interference from previous tasks, and a 'slow' policy that maintains strong cross-task generalization. The use of Bayesian estimation ensures an effective and unbiased selection between the fast and slow policies, enhancing the framework's adaptability and robustness. We also introduce a dual-reset mechanism and data relabeling to further enhance efficiency. Experimental

results demonstrate BFSF's effectiveness in resisting interference and show that it outperforms baseline methods in various non-stationary environments.

Although BFSF improves adaptability and efficiency in non-stationary MDPs, the current experiments are limited to a small set of environments. Other types of non-stationarity, such as blurred boundaries or stochastic non-stationary MDPs, have not yet been tested. Additionally, more realistic scenarios are needed to assess its applicability in real-world situations. In future work, we aim to develop more realistic benchmarks that align closely with real-world applications of non-stationary MDPs, while further testing BFSF and exploring new challenges.

## 6 Acknowledgment

This work was supported by AF Office of Scientific Research (AFOSR) under grant FA9550-25-1-0318 and with the generous support of the Amazon Research Award program.

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

## Technical Appendices and Supplementary Material

## A  About Theorem 2.1

### A.1  Proof of Theorem 2.1

*Proof.* First, since Bayesian estimation provides an unbiased estimate, the suboptimality of BFSF is bounded by the better performance between the fast and slow policies.

$$\text{Suboptimality}(BFSF) \leq \min(\text{Suboptimality}(Fast), \text{Suboptimality}(Slow)) \tag{4}$$

We analyze the suboptimalimality of the 'fast' policy $|\eta_M(\pi_M^*) - \eta_M(\pi_{M'}^*)|$, in the following proof.

Let MDP $M$ have the dynamic function $p$ and reward function $r$. Similarly, let MDP $M'$ have the dynamic function $p'$ and reward function $r'$. We denote $M_{p,r}$ as the MDP with dynamics function $p$ and reward function $r$. In Equation 5, we decompose $|\eta_M(\pi_M^*) - \eta_M(\pi_{M'}^*)|$ and use Theorem A.1, Lemma A.2, A.3 to complete the proof.

$$
\begin{aligned}
&|\eta_M(\pi_M^*) - \eta_M(\pi_{M'}^*)| \\
\leq &|\eta_M(\pi_M^*) - \eta_{M'}(\pi_{M'}^*)| + |\eta_{M'}(\pi_{M'}^*) - \eta_M(\pi_{M'}^*)| \\
= &|\eta_{M_{p,r}}(\pi_{M_{p,r}}^*) - \eta_{M_{p',r'}}(\pi_{M_{p',r'}}^*)| + |\eta_{M_{p',r'}}(\pi_{M_{p',r'}}^*) - \eta_{M_{p,r}}(\pi_{M_{p',r'}}^*)| \\
\leq &|\eta_{M_{p,r}}(\pi_{M_{p,r}}^*) - \eta_{M_{p',r'}}(\pi_{M_{p',r'}}^*)| \\
&+ |\eta_{M_{p',r'}}(\pi_{M_{p',r'}}^*) - \eta_{M_{p,r'}}(\pi_{M_{p',r'}}^*)| + |\eta_{M_{p,r'}}(\pi_{M_{p',r'}}^*) - \eta_{M_{p,r}}(\pi_{M_{p',r'}}^*)| \\
\leq &[\mathcal{D}_{\ell_1}(p_{M_1}, p_{M_2})(r_{max} + V_{max}) + r_{diff}]H + |U_{r_1,r_2}(\pi_{M_2}^*)| + \frac{1}{2}V_{max}\mathcal{D}_{\ell_1}(p_{M_2}(s,a), p_{M_1}(s,a))
\end{aligned}
\tag{5}
$$

$\square$

**Theorem A.1.** *(Relabeling gap 1)Let $M_1$, $M_2$ be two finite-horizon MDPs with the same reward function $r$. Then the distance of $\eta_{M_1}(\pi_{M_1}^*)$ and $\eta_{M_2}(\pi_{M_2}^*)$ is bounded by*

$$|\eta_{M_1}(\pi_{M_1}^*) - \eta_{M_2}(\pi_{M_2}^*)| \leq \epsilon_h, \tag{6}$$

*where $\epsilon_h = [\mathcal{D}_{\ell_1}(p_{M_1}, p_{M_2})(r_{max} + V_{max}) + r_{diff}](H - h), \forall h \in [H], s \in \mathcal{S}$.*

*Proof.* Since $\pi_{M_1}^*$ is the optimal policy of MDP $M_1$ and $\pi_{M_2}^*$ is the optimal policy of MDP $M_2$, $\eta_{M_1}(\pi_{M_1}^*) = V_{M_1}^*, \eta_{M_2}(\pi_{M_2}^*) = V_{M_2}^*$.

We begin our proof from the final horizon, $h = H$, and use the closeness at horizon $h$ to establish the closeness at horizon $h - 1$.

For the final horizon $h = H$, we have $V_{M,H}^*(s) = 0 \,\forall M$ since it is the terminal state. Therefore, $||V_{M_1,H}^*(s) - V_{M_2,H}^*(s)||_\infty \leq 0 = \epsilon_H$. Suppose

$$\forall s \in \mathcal{S}_h, ||V_{M_1,h}^*(s) - V_{M_2,h}^*(s)||_\infty \leq \epsilon_h. \tag{7}$$

We need to prove

$$\forall s \in \mathcal{S}_{h-1}, ||V_{M_1,h-1}^*(s) - V_{M_2,h-1}^*(s)||_\infty \leq \epsilon_{h-1}. \tag{8}$$

It is equivalent to prove

$$-\epsilon_{h-1} \leq V_{M_2,h-1}^*(s) - V_{M_1,h-1}^*(s) \leq \epsilon_{h-1}. \tag{9}$$

For simplicity, but without loss of generality, we will prove the inequality on the right-hand side of the above equation.

$$
\begin{aligned}
LHS =& \max_{a\in\mathcal{A}}\{\sum_{s'\in\mathcal{S}_h} p_{M_2}(s'|s,a)(r_2(s,a)+\gamma V^*_{M_2,h}(s'))\}\\
&- \max_{a\in\mathcal{A}}\{\sum_{s'\in\mathcal{S}_h} p_{M_1}(s'|s,a)(r_1(s,a)+\gamma V^*_{M_1,h}(s'))\}\\
=& \max_{a\in\mathcal{A}}\{\sum_{s'\in\mathcal{S}_h} [p_{M_1}(s'|s,a)(r(s,a)+\gamma V^*_{M_1,h}(s'))\\
&+ (p_{M_2}(s'|s,a)r_2(s,a)-p_{M_1}(s'|s,a)r_1(s,a)) + p_{M_2}(s'|s,a)V^*_{M_2,h}(s')\\
&- p_{M_1}(s'|s,a)V^*_{M_1,h}(s')]\} - \max_{a\in\mathcal{A}}\{\sum_{s'\in\mathcal{S}_h} p_{M_1}(s'|s,a)(r(s,a)+\gamma V^*_{M_1,h}(s'))\}\\
\leq& \max_{a\in\mathcal{A}}\{\sum_{s'\in\mathcal{S}_h} p_{M_1}(s'|s,a)(r(s,a)+\gamma V^*_{M_1,h}(s'))\}\\
&+ \max_{a\in\mathcal{A}}\{\sum_{s'\in\mathcal{S}_h} [(p_{M_2}(s'|s,a)r_2(s,a)-p_{M_1}(s'|s,a)r_1(s,a))\\
&+ p_{M_2}(s'|s,a)V^*_{M_2,h}(s')-p_{M_1}(s'|s,a)V^*_{M_1,h}(s')]\}\\
&- \max_{a\in\mathcal{A}}\{\sum_{s'\in\mathcal{S}_h} p_{M_1}(s'|s,a)(r(s,a)+\gamma V^*_{M_1,h}(s'))\}\\
=& \max_{a\in\mathcal{A}}\{\sum_{s'\in\mathcal{S}_h} [(p_{M_2}(s'|s,a)r_2(s,a)-p_{M_1}(s'|s,a)r_1(s,a))\\
&+ p_{M_2}(s'|s,a)V^*_{M_2,h}(s')-p_{M_1}(s'|s,a)V^*_{M_1,h}(s')]\}\\
=& \max_{a\in\mathcal{A}}\{\sum_{s'\in\mathcal{S}_h} [(p_{M_2}(s'|s,a)-p_{M_1}(s'|s,a))r_2(s,a) + \sum_{s'\in\mathcal{S}_h} p_{M_1}(s'|s,a)(r_2(s,a)-r_1(s,a))\\
&+ p_{M_2}(s'|s,a)V^*_{M_2,h}(s')-p_{M_1}(s'|s,a)V^*_{M_1,h}(s')]\}\\
\leq& \max_{a\in\mathcal{A}}\{\sum_{s'\in\mathcal{S}_h} |p_{M_2}(s'|s,a)-p_{M_1}(s'|s,a)|\}r_{max} + \max_{a\in\mathcal{A}}\{\sum_{s'\in\mathcal{S}_h} p_{M_1}(s'|s,a)r_{diff}\}\\
&+ \max_{a\in\mathcal{A}}\{\sum_{s'\in\mathcal{S}_h} (p_{M_2}(s'|s,a)-p_{M_1}(s'|s,a))V^*_{M_2,h}(s') + p_{M_1}(s'|s,a)(V^*_{M_2,h}(s')-V^*_{M_1,h}(s'))\}\\
\leq& \{D_{\ell_1}(p_{M_1}(\cdot|s,a),p_{M_2}(\cdot|s,a))\}(r_{max}+V_{max})+r_{diff}+1\cdot(V^*_{M_2,h}(s')-V^*_{M_1,h}(s'))\\
\leq& [D_{\ell_1}(p_{M_1},p_{M_2})(r_{max}+V_{max})+r_{diff}]+1\cdot\epsilon(h)\\
=& [D_{\ell_1}(p_{M_1},p_{M_2})(r_{max}+V_{max})+D_{\ell_1}(p_{M_1},p_{M_2})(r_{max}+V_{max})+r_{diff}](H-h)\\
=& [D_{\ell_1}(p_{M_1},p_{M_2})(r_{max}+V_{max})+r_{diff}](H-h+1).
\end{aligned}
$$
(10)

□

**Lemma A.2.** *Let $M_1, M_2$ be two MDPs with the same dynamics function, but different reward functions $r_1, r_2$. Define $U_{r_1,r_2}(\pi) = \mathbb{E}_{(s,a)\sim\rho^\pi_{M_1}}[r_2(s,a)-r_1(s,a)]$, which characterizes how erroneous the model is along trajectories induced by $\pi$. Then*

$$
\eta_{M_2}(\pi)-\eta_{M_1}(\pi) = U_{r_1,r_2}(\pi)
$$
(11)

*Proof.* We know that $\tilde{M}$ and $\hat{M}$ shares the same transition dynamics $p$, but different reward functions $\tilde{r}(s,a)=\hat{r}(s,a)-\lambda u(s,a)$. Therefore,

$$
\begin{aligned}
\eta_{M_2}(\pi) =& \mathbb{E}_{(s,a)\sim\rho^\pi_{M_1}}[r_2(s,a)]\\
=& \mathbb{E}_{(s,a)\sim\rho^\pi_{M_1}}[r_1(s,a)+(r_2(s,a)-r_1(s,a))]\\
=& \mathbb{E}_{(s,a)\sim\rho^\pi_{M_1}}r_1(s,a)-\mathbb{E}_{(s,a)\sim\rho^\pi_{M_1}}(r_2(s,a)-r_1(s,a))\\
=& \eta_{M_1}(\pi)+U_{r_1,r_2}(\pi).
\end{aligned}
$$
(12)

□

**Lemma A.3.** *(Telescoping lemma) [36, 32]. Let $M_1$ and $M_2$ be two MDPs with the same reward $r(s, a)$, but different dynamics $p_{M_1}$ and $p_{M_2}$ respectively. Let*

$$
\begin{aligned}
G_{M_2}{}^{\pi}(s, a) := \\
\mathbb{E}_{s' \sim p_{M_2}(s,a)}[V_{M_1}^{\pi}(s')] - \mathbb{E}_{s' \sim p_{M_1}(s,a)}[V_{M_1}^{\pi}(s')],
\end{aligned}
\tag{13}
$$

*Then,*

$$
\eta_{M_2}(\pi) - \eta_{M_1}(\pi) = \gamma \mathbb{E}_{(s,a) \sim \rho_{M_2}^{\pi}}[G_{M_2}^{\pi}(s, a)].
\tag{14}
$$

*For each $s \in \mathcal{S}, a \in \mathcal{A}$, a $\ell_1$-based bound of $|G_{M_2}^{\pi}(s, a)|$ is*

$$
|G_{M_2}^{\pi}(s, a)| \leq \frac{1}{2} V_{max} \delta_{\ell_1}(p_{M_2}(s, a), p_{M_1}(s, a)).
\tag{15}
$$

## A.2 Bound Considering the Optimization Error

Theorem 2.1 accounts for the error introduced by the relabeling process. To maintain consistency with prior work, we explicitly incorporate optimization suboptimality, following the approach in [34], by considering the policy obtained after $K$ policy-update iterations $\pi_{M'}^K$, rather than the idealized optimal policy $\pi_{M'}^*$.

$$
\begin{aligned}
&\text{Total Suboptimality} \\
\leq& |\eta_M(\pi_M^*) - \eta_M(\pi_{M'}^K)| \\
\leq& |\eta_M(\pi_M^*) - \eta_{M'}(\pi_{M'}^*)| + |\eta_{M'}(\pi_{M'}^*) - \eta_{M'}(\pi_{M'}^K)| + |\eta_{M'}(\pi_{M'}^K) - \eta_M(\pi_{M'}^K)| \\
\leq& [D_1(p_M, p_{M'})(r_{\max} + V_{\max}) + r_{\text{diff}}]H + |U_{r,r'}(\pi_{M'}^*)| + \frac{1}{2} V_{\max} D_1(p_{M'}(s, a), p_M(s, a)) \\
&+ C \cdot \frac{\phi_{\mu,\sigma} \cdot \gamma}{(1 - \gamma)^2} \cdot |A| \cdot (\log n)^{1+2\xi^*} \cdot n^{(\alpha^* - 1)/2} + \frac{4\gamma^{K+1}}{(1 - \gamma)^2} \cdot R_{\max},
\end{aligned}
\tag{16}
$$

where each term explicitly captures different sources of error:

- The **first 3 terms** (matching our original Theorem 3.1) represent the suboptimality caused by the **relabeling process**, quantifying the error introduced due to differences in dynamics and reward functions between the original MDP $M$ and the relabeled MDP $M'$.

- The newly introduced **4th and 5th terms** explicitly quantify the **optimization suboptimality**, representing errors arising from finite-sample approximations and iterative optimization.

# B Experiments

## B.1 Implementation Details

The experiments are repeated three times, with the mean and standard deviation shown in the curves and table.

The common hyperparameters are consistent with the original PEARL implementation. Additionally, the context consists of 200 episodes, and the relabeling percentage is set to $50\%$ (i.e., half of the used batch is relabeled). The window of recent data $w = 100$ episodes. The reset frequency $\nu = 50$ episodes. The discount factor $\gamma = 0.99$.

## B.2 Full Learning Curves

A complete comparison of the learning curves for all four baselines is provided in Figure 6.

## B.3 Sensitive Studies and Other Experiments

**Sensitive study on the relabeling percentage** We performed an sensitive study on the relabeling percentage in the Table 2.

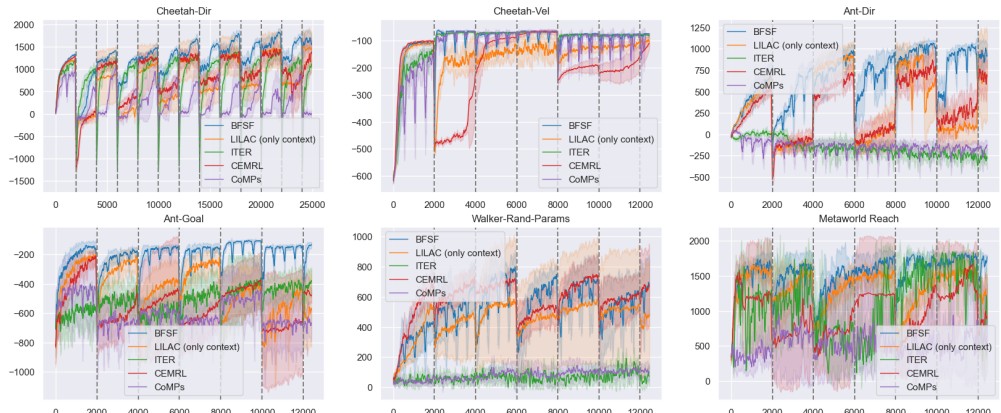

Figure 6: The learning curve of BFSF with all the baselines including LILAC, ITER, CEMRL and CoMPs.

| Relabel Percentage (%) | 0 | 10 | 20 | 30 | 40 | 50 |
|---|---|---|---|---|---|---|
| Average Return | 1055.5 | 954.9 | 1046.3 | 1097.5 | 1186.4 | 1039.4 |

| Relabel Percentage (%) | 60 | 70 | 80 | 90 | 100 |
|---|---|---|---|---|---|
| Average Return | 1217.7 | 1200.9 | 1148.1 | 1027.6 | 1129.1 |

Table 2: Sensitive study on the relabeling percentage

**Sensitive study on the window of recent data $w$ and the reset frequency $\nu$** is shown in Table 3. For reference, the average return of the baseline (LILAC) is 757.8. The results indicate that performance of our method is relatively insensitive to the choice of window size $w$ and reset frequency $\nu$.

**The experiment in gradually changing environments** To validate our method in dynamically changing environments, we conducted supplementary experiments in Table 4 in a 180-task gradually changing Ant-Goal environment, with goals evenly distributed around a circle.

**The experiment in stochastically changing environments** As to settings with highly stochastic task boundaries, we conduct experiments on a Cheetah-Dir environment in Table 5. This environment consists of two tasks: moving forward and moving backward. A boundary period (one-third of the task length) exists during task switching, with each task having a probability of $50\%$.

## C Algorithm Details

### C.1 Data Relabeling

To elaborate, relabeling is accomplished by a learned dynamics and reward model $s', r = f(s, a)$ [15], which estimates the next state $s'$ and reward $r$ after taking action $a$ in the state $s$. We can

| | $w = 10$ | $w = 50$ | $w = 100$ | $w = 150$ | $w = 200$ |
|---|---|---|---|---|---|
| BFSF | 1163.3 | 1273.6 | 1209.0 | 1121.1 | 945.5 |

| | $\nu = 10$ | $\nu = 25$ | $\nu = 50$ | $\nu = 75$ | $\nu = 100$ |
|---|---|---|---|---|---|
| BFSF | 1250.6 | 1169.0 | 1209.0 | 1132.1 | 1116.6 |

Table 3: Sensitive study on the window of recent data $w$ and the reset frequency $\nu$.

|  | BFSF | Baseline: LILAC |
|---|---|---|
| Gradual Ant-Goal | $-328.5 \pm 2.1$ | $-619.2 \pm 3.7$ |

Table 4: The experiment results in a gradual Ant-Goal environment. BFSF outperforms the baseline, showing its ability to adapt continuously to a changing environment.

|  | BFSF | Baseline: LILAC |
|---|---|---|
| Stochastic Ant-Goal | $887.8 \pm 21.6$ | $615.4 \pm 5.7$ |

Table 5: The experiment results in a stochastic Ant-Goal environment. BFSF significantly outperforms baseline LILAC in such an environment.

substitute the original next state and reward in the experience replay, even from different tasks, with those predicted by the model, represented as $s'_{relabel}, r_{relabel} = f(s, a)$.

Relabeling is widely used in the RL community [29, 33]. The underlying principle is to maximize data reuse for sample efficiency. In our work, given that the environment evolves over time, we leverage a context-based dynamics model $f(s, a, c)$, which provides different dynamics depending on context $c$.

As discussed in [29, 33] and confirmed by our ablation studies, data relabeling substantially enhances sample efficiency. In our work, we adopt data relabeling to mitigate the issue of performance degradation in a non-stationary environment, because the increased amount of relabeled data allow the 'fast' policy to better and faster adapt to these tasks, which is verified by our ablation results.

### C.2    Motivation for the Dual-Reset Mechanism

The dual-reset mechanism was introduced upon observing that RL algorithms tend to encounter performance degradation when alternating between tasks, a trend corroborated by prior research [4]. This mechanism effectively addresses the issue. We present the phase performance for SAC in the alternating Cheetah-Dir environment (where the two tasks are moving forward and backward) in Table 6.

### C.3    Bayesian Fast-Slow Framework

We only utilize data within a recent window, which keeps the effective sample size small, and we set the prior value to a dynamically updated upper bound to encourage exploration. As a result, if a policy has not been sufficiently selected, the prior strongly influences the Bayesian estimate, leading to a large posterior value that naturally encourages exploration of that policy.

| Phase performance | Task 1 | Task 2 | Task 1 | Task 2 | Task 1 |
|---|---|---|---|---|---|
| Without dual-reset | 954.6 | 117.8 | 92.0 | 8.6 | $-27.4$ |
| With dual-reset | 759.7 | 625.6 | 697.5 | 409.8 | 782.7 |

Table 6: In the first phase (Task 1), SAC without the dual-reset mechanism performs well, even outpacing the SAC with dual-reset. However, during the alternating tasks, the performance degrades significantly.

## D    Posterior Calculation of Normal Distributions

Assume $\phi$ is known.

$$
\begin{aligned}
p(\mu|\{R_{i_1}, R_{i_2}, \cdots\}, 1/\phi) &\propto p(\mu)p(\{R_{i_1}, R_{i_2}, \cdots\}|\mu, 1/\phi) \\
&\propto \exp\big\{ -\frac{\phi_0}{2}(\mu - \mu_0)^2 \big\} \times \exp\big\{ -\frac{n\phi}{2}(\mu - \overline{y})^2 \big\} \\
&\propto \exp\big\{ -\frac{1}{2}(\phi_0 + n\phi)\mu^2 + \frac{1}{2}(2\mu_0\phi_0 + 2n\phi\overline{y})\mu \big\} \\
&\propto \exp\big\{ -\frac{1}{2}(\phi_0 + n\phi)(\mu - \frac{\phi_0\mu_0 + n\phi\overline{y}}{\phi_0 + n\phi})^2 \big\} \\
&\sim \text{Normal}(\mu_1, \sigma_1^2), \\
\text{where } \mu_1 &= \frac{\phi_0\mu_0 + n\phi\overline{R}}{\phi_0 + n\phi}, \sigma_1^2 = \frac{1}{\phi_0 + n\phi}.
\end{aligned}
\tag{17}
$$

## E    Visualization of Choosing Slow/Fast Policies

We presented visualization in left sub-figure of Figure 5 in our paper (attached here in Figure 7). In the left sub-figure, the 'fast' policy predominates in the selection during the initial phases, while the 'slow' policy shows competitive performance as the amount of accumulated data from different tasks increases. Additionally, in the latter part of a single phase, the 'fast' policy surpasses the 'slow' policy after learning from relabeled recent data.

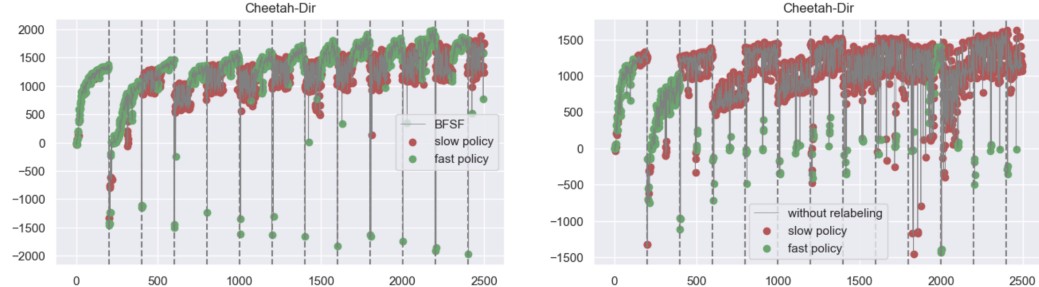

Figure 7: A comparison illustrating the performance of the 'slow' and 'fast' policies. The main difference is that with relabeling, the performance of the 'fast' policy remains higher throughout the phases, rather than significantly dropping after the initial phases.

