# OpenReview forum: "A Bayesian Fast-Slow Framework to Mitigate Interference in Non-Stationary Reinforcement Learning"
_NeurIPS.cc/2025/Conference — NeurIPS 2025 poster_

### Official Review · Reviewer_DwXB · 2025-07-03

**Clarity:** 2
**Significance:** 2
**Originality:** 3
**Rating:** 5
**Confidence:** 4

**Summary:**

This paper proposes an RL framework (BFSF) for continual non-stationary reinforcement learning. The agent maintains two policies: a “slow” meta-RL policy (implemented with PEARL) that exploits long-term experience for cross-task generalisation, and a “fast” policy trained only on recent data to react quickly to distribution shifts and reduce negative transfer (termed “interference”). The current policy is chosen each epoch via Bayesian model selection on a sliding window of recent returns. The fast policy further uses (i) a dual-reset mechanism that periodically re-initialises one of two identical networks to avoid performance collapse, and (ii) context-conditioned data relabeling to augment its recent replay buffer. Theoretical analysis gives a sub-optimality bound that decomposes errors from relabelling and optimisation. Experiments on six tasks (from MuJoCo and Meta-World benchmarks) plus two “infinite-world” variants show improved average returns over baselines such as LILAC, ITER, CEMRL and CoMPs.

**Questions:**

Please refer to the weaknesses above. Mainly,
- Is data relabeling also used for all the baselines with fast policies? If not, please do so.
- Report PEARL alone (the "slow" learner from prior work). Also report the "fast" learner used in Algorithm 2 step 10 with experience relabeling. The absence of these two baselines are of major concern to me.
- Increase seeds to at least 10 and include confidence intervals. Otherwise the claimed gains may not be significant.
- Please define all symbols in Theorem 2.1 and provide a proof sketch. Under what realistic conditions can the bound be < V_max? If the bound is always loose, consider reframing the theory section.
- Explain algorithmic timing. Describe precisely how training, dual-reset, and environment interaction interleave.
- In Fig. 2, what causes the periodic drops in BFSF between switches, and why are the drops absent at 6 k / 10 k steps on Meta-World Reach?
-  Line 497 says "As we cannot include images in this rebuttal" in the appendix of the submission. Please include said images.

**Ethical Concerns:**

["NO or VERY MINOR ethics concerns only"]

**Final Justification:**

The authors really took some time and effort to properly address my concerns and provide a very detailed response. I am happy with their clarification and happy to increase my score to an accept accordingly.

**Limitations:**

There is no broader impacts statement.

**Paper Formatting Concerns:**

There is no introduction section title and no broader impacts statement.

**Quality:**

2

**Strengths And Weaknesses:**

# Strengths

### Quality
- Really solid engineering effort. BFSF combines a principled fast/slow decomposition, Bayesian model selection, dual-reset, and experience relabeling using learned world model. Each component is at least intuitively motivated and evaluated in ablations.
- It outperforms strong baselines across eight non-stationary tasks and maintains higher returns through task switches.
- The empirical investigation of “interference” is interesting, especially with the visualisations in Figure 3.
- The hyperparameter sensitivity ablations are also very commendable.

### Clarity
- The High-level intuition of BFSF is easy to grasp and the algorithms are summarised in pseudo-code.
- Decent related-work section clearly positions BFSF with respect to meta-RL and continual RL literature.

### Significance
- Addresses an important pain-point in deploying meta-RL agents in streaming settings where sharp task boundaries are unknown.
- The fast/slow idea could inspire follow-up work that plugs in different meta-learners or fast learners, and the infinite-world experiments push beyond the usual episodic benchmarks.


### Originality
It provides a novel combination of known components. While fast-adaptation, Bayesian policy selection, and relabeling are known ingredients, their integration as a coherent framework for mitigating cross-task interference in non-stationary MDPs is new. The dual-reset mechanism is also a simple but clever trick that may interest practitioners suffering from representation collapse (although it is unclear how sensitive it is to its hyperparameter).


# Weaknesses

### Quality
- Theoretical section is unfortunately weak. Theorem 2.1 appears vacuous if I understand it correctly. The bound is ≥ V_max when M≠M′, with undefined symbols p_M, η_M, etc. and the promised proof sketch (line 145) is never given.
- There is no discounting in the MDP problem definition. Is there no discounting in the experiments? In general, critical hyper-parameters like discount factor, window size w, reset frequency ν, RL algorithm for line 10 in Alg. 2) and number of seeds (only 3) are either buried in the appendix or absent, undermining statistical reliability.
- PEARL is used as the slow policy but is not reported as a standalone baseline. Likewise there is no “fast learner + relabeling only” baseline (since that's not new to this work), so improvements may conflate benefits from the proposed approach.
- It looks like the authors did not also do data relabeling for all the baselines with fast policies (e.g. ITER). This is a major concern if that is indeed the case.
- Unlike the authors claim on line 471, Table 3 shows (I assume average returns) that the approach is highly sensitive to the window size w and reset frequency v with a very clear trend (higher w and v lead to poorer performance).
- For Figure 4.b, the authors say "without fast policy" is identical to LILAC. Shouldn't it be PEARL since that's what the authors claim to be using for the slow policy? There's also too much noise in the figure. E.g. It is hard to justify the "Dual-Reset Ablation" interpretation given by the authors given the noise. An easier domain like Cheetah-vel or Ant-Goal (from looking at the results in Fig 2) would have been better for this ablation.
- The Bayesian inference ablation is inconclusive. Figure 4(b) shows almost identical returns on average with and without Bayesian selection, casting doubt on the necessity of the Bayesian component.


### Clarity
- Key quantities like Posterior(...), p_M, η_M, are undefined in the main text.
- Several typos hinder readability (lines 114, 122, 228).
- Figures use raster graphics and colour palettes that are hard to distinguish for colour-blind readers;
- Regular drops in BFSF curves in-between task switches are unexplained (e.g. in Fig 2). Also why does BFSF have no performance drop at 6000 and 10000 in Metaworld Reach (Fig 2)?
- Algorithmic flow between Alg. 1, Alg. 2, and environment interaction is still ambiguous: when exactly is the environment reset, and how are online interactions interleaved with dual-reset?

### Significance and Originality
- With only three seeds it is unclear if the results are significant.
- The main novelty in this work seems to be from the specific Bayesian inference and dual-reset ideas. Unfortunately, Figure 4 does not support the claim that these are useful significant components of the approach.

Overall, the paper tackles an important practical issue and shows promising empirical gains. However, clarity, baseline coverage, and theoretical rigor need strengthening before the contribution can be considered fully conclusive.

---

> ### Author Rebuttal · Authors · 2025-07-31
>
> Thanks for the valuable review.
>
> **Q1** Is data relabeling also used for all the baselines with fast policies? If not, please do so.
>
> **A1** Thank you for the valuable suggestion. We will include baselines with data relabeling in the final version. Due to time constraints, we currently provide a comparison between our algorithm without relabeling and the baselines:
>
> ||BFSF w/o relabeling|LILAC|ITER|CEMRL|CoMPs|
> |----|----|----|----|----|----|
> |Cheetah-Dir|$1053.9\pm55.0$|$757.7\pm94.9$|$813.4\pm19.0$|$852.6\pm36.1$|$277.3\pm40.2$|
>
> **Q2, W3, W4, W6** Report PEARL alone (the "slow" learner from prior work). Also report the "fast" learner used in Algorithm 2 step 10 with experience relabeling. The absence of these two baselines are of major concern to me.
>
> **A2** LILAC, which we use as one of our baselines, is just the non-stationary version of PEARL. PEARL is a meta-RL algorithm that requires the task id, which is not available in the non-stationary problem setting. LILAC transfers PEARL into the non-stationary MDPs scenario.
>
> Another baseline, ITER, can be viewed as the ‘fast’ learner only, as ITER also adopts the mechanism of periodically resetting the model.
>
> Based on the comparisons with these baselines, our method demonstrates strong performance, addressing concerns regarding the adequacy of baseline coverage. Nonetheless, we recognize the importance of clearer presentation and will improve the baseline annotations and related discussions in the next revision.
>
> **Q3, W13** Increase seeds to at least 10 and include confidence intervals. Otherwise the claimed gains may not be significant.
>
> **A3** Thank you for the helpful advice. We will increase the number of seeds in the next version. For now, we include BFSF results with three additional seeds:
>
> ||Cheetah-Dir|Cheetah-Vel|Ant-Dir|Ant-Goal|
> |----|----|----|----|----|
> |BFSF|$1189.4\pm41.5$|$-97.8\pm2.8$|$574.9\pm225.9$|$-369.1\pm208.1$|
> |LILAC|$757.7\pm94.9$|$-154.6\pm2.5$|$318.5\pm66.2$|$-426.6\pm4.6$|
>
> **Q4, W1** Please define all symbols in Theorem 2.1 and provide a proof sketch. Under what realistic conditions can the bound be < V_max? If the bound is always loose, consider reframing the theory section.
>
> **A4** The symbol definitions appear on Line 142; we reproduce and briefly explain them below:
> - $M,M’$: the original and relabeled MDPs.
> - $H$: the horizon.
> - $r_{max},V_{max}$: the maximum reward and value.
> - $r_{diff}$: the maximum reward gap between $M,M’$.
> - $U_{r,r'}(\pi)=E_{(s,a)\sim\rho^\pi_{M}}[r’(s,a)-r(s,a)]$: the average reward difference between two MDPs under the same occupancy measure $\rho_M^{\pi}$.
> - $D_{\ell_{1}}$: the L1 distance.
>
> **Proof Sketch** The bound decomposes the suboptimality into two parts that bridge the true MDP $M$ and the relabeled MDP $M'$:
> 1. $|\eta_M(\pi^\ast_M)-\eta_{M’}(\pi^\ast_{M’})|$: the difference between the optimal returns in $M$ and $M’$, corresponding to the first RHS term.
> 2. $|\eta_{M’}(\pi^\ast_{M’})-\eta_{M}(\pi^\ast_{M’})|$: the return difference when applying the same policy $\pi_{M’}^\ast$ under two different MDPs, $M$ and $M'$. This term can be further decomposed into two parts, corresponding to the second and third terms in Theorem 2.1, because the MDPs differ in both reward and dynamics.
>
> **Tightness:** The bound is not as large as $V_{max}$, as $V_{max}$ can be viewed as a constant multiplied by an L1 distance (e.g. $V_{max}D_{\ell_1}(p_{M’}(s,a),p_M(s,a))$). The relabeled MDP $M’$ trained to closely approximate the original MDP $M$, so the L1 distance tends to be small after training.
>
> **Q5, W12** Explain algorithmic timing. Describe precisely how training, dual-reset, and environment interaction interleave.
>
> **A5** In each iteration, we first select either the ‘fast’ or ‘slow’ policy. We then collect 5 episodes, with each episode consisting of 200 steps, through environment interaction. Afterward, we perform 2000 steps of RL training. The model reset occurs every 50 episodes.
>
> **Q6, W11** In Fig. 2, what causes the periodic drops in BFSF between switches, and why are the drops absent at 6 k / 10 k steps on Meta-World Reach?
>
> **A6** The periodic drops are caused by abrupt task changes with potentially conflicting objectives — for example, in Cheetah-Dir, the tasks involve moving left versus right. In contrast, in Meta-World Reach, although goals are randomly sampled across the space, neighboring tasks may occasionally result in nearby goals, leading to smoother transitions and less pronounced performance drops.
>
> **Q7** Line 497 says "As we cannot include images in this rebuttal" in the appendix of the submission. Please include said images.
>
> **A7** We apologize for the missing image. Since images cannot be included in the rebuttal itself, we will ensure that the final version of the paper includes the missing figure with the performance curves.
>
> **W2** There is no discounting in the MDP problem definition. Is there no discounting in the experiments? In general, critical hyper-parameters like discount factor, window size w, reset frequency ν, RL algorithm for line 10 in Alg. 2) and number of seeds (only 3) are either buried in the appendix or absent, undermining statistical reliability.
>
> **Answer:** Thank you for pointing this out. We will make sure to explicitly present these key hyperparameters alongside the main definitions in future versions for better clarity. Below are the values used in our experiments:
>
> - Discount factor: $0.99$. The MDP definition omits discounting since our environments are finite-horizon tasks where discounting is not strictly necessary. However, we use standard discounted return during training.
>
> - Window size $w$: $100$.
>
> - Reset frequency $\nu$: $50$.
>
> **W5** Unlike the authors claim on line 471, Table 3 shows (I assume average returns) that the approach is highly sensitive to the window size w and reset frequency v with a very clear trend (higher w and v lead to poorer performance).
>
> **Answer:** We agree that the performance varies across different choices of $w$ and $\nu$, as shown in Table 3. However, even under suboptimal hyperparameters, our approach consistently outperforms the baseline (LILAC average return: 757.8), demonstrating the robustness and effectiveness of BFSF.
>
> |  | $w=10$ | $w=50$ | $w=100$ | $w=150$ | $w=200$ |
> | ---- | ---- | ---- | ---- | ---- | ---- |
> | BFSF | $1163.3$ | $1273.6$ | $1209.0$ | $1121.1$ | $945.5$ |
>
> ||$\nu=10$|$\nu=25$|$\nu=50$ | $\nu=75$ | $\nu=100$ |
> |----|---- | ---- | ---- | ---- | ---- |
> |BFSF|$1250.6$ | $1169.0$ | $1209.0$ | $1132.1$ | $1116.6$ |
>
> **W6** There's also too much noise in the figure. E.g. It is hard to justify the "Dual-Reset Ablation" interpretation given by the authors given the noise. An easier domain like Cheetah-vel or Ant-Goal (from looking at the results in Fig 2) would have been better for this ablation.
>
> **Answer:** Thank you for the suggestion. We agree that domains like Cheetah-vel or Ant-goal may yield cleaner curves and help highlight component effects. We chose Cheetah-dir because it presents a more challenging setting where interference is more severe, making the benefits of our design more evident. That said, we appreciate the reviewer’s feedback and will consider ways to improve the clarity and readability of the curves in future revisions.
>
> **W14** The main novelty in this work seems to be from the specific Bayesian inference and dual-reset ideas. Unfortunately, Figure 4 does not support the claim that these are useful significant components of the approach.
>
> **Answer:** Thank you for the comment. While Bayesian inference and the dual-reset mechanism are indeed important components of our method, we would like to clarify that the core novelty lies in the dynamic switching between the fast and slow policies, which directly addresses the interference problem in non-stationary environments. The dual-reset and data relabeling mechanisms help these modules function effectively, while Bayesian inference provides a theoretically grounded and adaptive way to select between them.
>
> More specifically, compared to the ablations without dual-reset or Bayesian inference, BFSF demonstrates a clear advantage at the beginning of the third phase: these ablated versions struggle to leverage experience from the first phase due to interference, while BFSF maintains better policy continuity. This suggests that the full BFSF framework, including Bayesian switching and dual resets, contributes meaningfully to mitigating interference.
>
> **W8-12** Weakness in clarity.
>
> **Answer:** Thank you for the careful reading and constructive feedback. We will correct all typographical errors and improve figure clarity in the final version.

---

> > ### Author Response · Authors · 2025-08-08
> >
> > Dear Reviewer DwXB,
> >
> > Thank you very much for your detailed and thoughtful review. We have carefully addressed all of your comments and concerns in our rebuttal. If you have any further questions or require additional clarifications, we would greatly appreciate your feedback.
> > Your follow-up response would be extremely valuable to ensure a fair and accurate evaluation of our work.
> >
> > Thank you again for your time and consideration.
> >
> > Best regards,
> >
> > The Authors

---

> > > ### Comment · Reviewer_DwXB · 2025-08-09
> > > **Thanks**
> > >
> > > Thank you to the authors for really taking the time and effort to properly address my concerns and provide a very detailed response. I am happy with their clarifications.

---

### Official Review · Reviewer_tpda · 2025-07-03

**Clarity:** 3
**Significance:** 3
**Originality:** 3
**Rating:** 4
**Confidence:** 4

**Summary:**

This paper introduces the Bayesian Fast-Slow Framework (BFSF), a novel approach to address the challenge of interference in non-stationary reinforcement learning environments. Unlike previous works that focus solely on cross-task generalization, BFSF dynamically switches between a fast policy trained on recent data for rapid adaptation and a slow policy trained via meta-RL to leverage past experience. A Bayesian inference mechanism determines the optimal policy to use at each point in time.

**Questions:**

- Could the Bayesian selection mechanism be enhanced with exploration to avoid early commitment bias? The current policy selection is fully deterministic based on posterior mean estimates. This may lead to under-utilization of one policy (e.g., the slow policy) if the other dominates early on. Have you considered incorporating stochastic policy selection mechanisms to encourage balanced exploration of both fast and slow policies, especially in the early phases?
- How robust is the fast policy’s performance to inaccuracies in the context-based dynamics model used for relabeling? Since relabeling heavily depends on the quality of the learned model, it would be helpful to understand how model errors affect fast policy training. Have you evaluated the impact of model approximation errors on policy performance?
- What happens when fast policy dominates data collection for extended periods? Although slow policy is updated in each epoch, it depends on data collected through interaction through the past policy. It would be helpful to clarify how such imbalance is mitigated in practice, and whether you considered mechanisms for guaranteeing minimum data exposure for each policy.
- In Figure 4(b), the ablation results suggest that the fast policy and relabeling have the most significant impact on overall performance, whereas the removal of Bayesian inference seems to cause only a minor degradation. Could you clarify the relative importance of the Bayesian selection mechanism in the overall framework? Specifically, how critical is Bayesian inference to the stability and adaptability of BFSF? If the system performs almost as well without Bayesian inference and slow policy, what unique benefits does it provide in more challenging or uncertain settings?

**Ethical Concerns:**

["NO or VERY MINOR ethics concerns only"]

**Final Justification:**

First, I sincerely apologize for the delayed response. I had intended to submit my updated review during an overseas trip, but it appears that the submission did not go through, most likely due to unstable airport network conditions.

Regarding the paper, after carefully reading the authors’ rebuttal, I found their responses sufficiently address all of my initial concerns.
Given the clarifications provided, I have updated my evaluation to better reflect the quality and contribution of the work.

**Limitations:**

Yes

**Paper Formatting Concerns:**

1. The submitted paper content: No issue
2. Paper references: No issue
3. The NeurIPS paper checklist: No issue

**Quality:**

3

**Strengths And Weaknesses:**

1. Strengths
- The paper introduces a well-motivated and novel framework (BFSF) specifically designed to tackle interference in non-stationary RL environments.
- The use of Bayesian inference for dynamically selecting between fast and slow policies is principled and adaptive, allowing the agent to respond to changing task distributions efficiently.
- The incorporation of the dual-reset mechanism and context-aware data relabeling significantly enhances the fast policy’s adaptability and sample efficiency.

2. Weaknesses
- The framework deterministically selects the policy (fast or slow) with the higher expected return, without incorporating stochastic exploration, which may lead to overcommitting to suboptimal policies in early stages.
- If one policy (e.g., fast) dominates selection early on, the other (e.g., slow) may suffer from under-training due to limited data collection, which is not fully analyzed in the paper.
- The performance of the fast policy partially depends on the accuracy of the learned context-based dynamics model, but the paper does not deeply analyze its limitations or robustness.
- The framework introduces several new hyperparameters (e.g., reset frequency, window size for Bayesian updates), but sensitivity to these is not extensively studied or theoretically justified.

---

> ### Author Rebuttal · Authors · 2025-07-31
>
> Thanks for the valuable review.
>
> **Q1, W1** Could the Bayesian selection mechanism be enhanced with exploration to avoid early commitment bias? The current policy selection is fully deterministic based on posterior mean estimates. This may lead to under-utilization of one policy (e.g., the ‘slow’ policy) if the other dominates early on. Have you considered incorporating stochastic policy selection mechanisms to encourage balanced exploration of both ‘fast’ and ‘slow’ policies, especially in the early phases?
>
> **A1** Exploration is inherently incorporated in our approach. Specifically, we only utilize data within a recent window, which keeps the effective sample size small, and we set the prior value $\mu_0$ to a dynamically updated upper bound to encourage exploration. As a result, if a policy has not been sufficiently selected, the prior strongly influences the Bayesian estimate, leading to a large posterior value that naturally encourages exploration of that policy.
>
> Moreover, the ‘slow’ and ‘fast’ policies do not compete exclusively as in standard bandit settings. The experience gathered by one policy can be efficiently leveraged by the other. Evidence for this can be seen in existing offline meta-RL research [1], where even fully offline datasets are sufficient to support effective context-based policies.
>
> Lastly, our algorithm selects between the ‘fast’ and ‘slow’ policies by sampling from their respective posterior distributions and choosing based on the probability that one sample exceeds the other, which inherently provides exploration capability.
>
> **Q2, W3** How robust is the ‘fast’ policy’s performance to inaccuracies in the context-based dynamics model used for relabeling? Since relabeling heavily depends on the quality of the learned model, it would be helpful to understand how model errors affect ‘fast’ policy training. Have you evaluated the impact of model approximation errors on policy performance?
>
> **A2** Thank you for the question. The context-based dynamics model is used to relabel past experiences to make them more relevant to the current task, thereby accelerating the fast policy’s adaptation. When the model is inaccurate, the relabeled data may become less helpful, but the fast policy can still rely on newly collected (on-policy) data. As a result, the impact of model errors is typically limited to short-term adaptation performance, and does not severely affect long-term learning.
>
> To evaluate the sensitivity to model quality, we conducted an ablation study on the relabeling percentage (Appendix B.3), summarized below:
>
> |Relabel Percentage (%)|0|10|20|30|40|50|60|70|80|90|100|
> |----|----|----|----|----|----|----|----|----|----|----|----|
> |Average Return|1055.5|954.9|1046.3|1097.5|1186.4|1039.4|1217.7|1200.9|1148.1|1027.6|1129.1|
>
> As the results show, our algorithm maintains relatively stable performance across a wide range of relabeling percentages, demonstrating that our method is robust and does not heavily depend on precise tuning.
>
> **Q3, W2** What happens when fast policy dominates data collection for extended periods? Although slow policy is updated in each epoch, it depends on data collected through interaction through the past policy. It would be helpful to clarify how such imbalance is mitigated in practice, and whether you considered mechanisms for guaranteeing minimum data exposure for each policy.
>
> **A3** This concern relates closely to the explanation in **A1**: the ‘slow’ and ‘fast’ policies do not compete exclusively, and the experience collected by one policy can be effectively leveraged by the other.
>
> To better answer your question, we can analyze how the ‘fast’ and ‘slow’ policy functions in a real experiment in Figure 5. In the beginning, ‘fast’ policy dominates as there are not enough multi-task data for ‘slow’ policy to generalize, while in the tasks afterward, the ‘slow’ context-based policy serves as a good initial point and the ‘fast’ policy further improves efficiently. To summarize, the ‘slow’ policy provides reasonable performance when encountering a new task, while the ‘fast’ policy dominates the improvement inside a task, which can also be utilized by the ’slow’ policy.
>
> **Q4, W7** In Figure 4(b), the ablation results suggest that the fast policy and relabeling have the most significant impact on overall performance, whereas the removal of Bayesian inference seems to cause only a minor degradation. Could you clarify the relative importance of the Bayesian selection mechanism in the overall framework? Specifically, how critical is Bayesian inference to the stability and adaptability of BFSF? If the system performs almost as well without Bayesian inference and slow policy, what unique benefits does it provide in more challenging or uncertain settings?
>
> **A4** We appreciate the reviewer’s insightful comment. As shown in Figure 4(b), the fast policy and relabeling indeed have the most direct and substantial impact on performance, with the removal of Bayesian inference resulting in only a minor drop in performance under the current benchmarks. However, it’s important to note that the ablation without Bayesian inference can be viewed as a deterministic approximation of the Bayesian posterior, using the mean. The full Bayesian approach still provides a principled and theoretically grounded framework for decision-making.
>
> While the fast policy and relabeling can improve performance independently in simpler tasks or when prior knowledge is strong, the true strength of Bayesian inference shines in more complex and uncertain environments. In such settings, Bayesian reasoning helps mitigate early commitment biases, improving the agent’s ability to adapt over time. By weighing policies based on their long-term effectiveness, Bayesian inference ensures greater stability and robustness, especially in non-stationary or noisy conditions.
>
> **W4** The framework introduces several new hyperparameters (e.g., reset frequency, window size for Bayesian updates), but sensitivity to these is not extensively studied or theoretically justified.
>
> **Answer:** Appendix B.3 provides a sensitivity study on the window of recent data $w$ and the reset frequency $\nu$. Additionally, we present the following study for reference:
>
> **Sensitivity Study on the window of recent data $w$:** (For reference, the average return of the baseline (LILAC) is $757.8$.)
>
> |  | $w=10$ | $w=50$ | $w=100$ | $w=150$ | $w=200$ |
> | ---- | ---- | ---- | ---- | ---- | ---- |
> | BFSF | $1163.3$ | $1273.6$ | $1209.0$ | $1121.1$ | $945.5$ |
>
> **Sensitivity Study on the reset frequency $\nu$:**
>
> ||$\nu=10$|$\nu=25$|$\nu=50$ | $\nu=75$ | $\nu=100$ |
> |----|---- | ---- | ---- | ---- | ---- |
> |BFSF|$1250.6$ | $1169.0$ | $1209.0$ | $1132.1$ | $1116.6$ |
>
> In the Bayesian update, the prior value $\mu_0$ is set to a dynamically updated upper bound to encourage exploration. The value of $\phi_0$ is set to 1 without further tuning.
>
>
> [1] Kate et. al., Efficient Off-Policy Meta-Reinforcement Learning via Probabilistic Context Variables. ICML 2019.

---

### Official Review · Reviewer_fHvL · 2025-07-03

**Clarity:** 4
**Significance:** 3
**Originality:** 3
**Rating:** 5
**Confidence:** 4

**Summary:**

This paper introduces the Bayesian Fast-Slow Framework (BFSF) to address the problem of interference in non-stationary reinforcement learning environments. The key issue is that when RL agents encounter changing tasks over time, learning from previous tasks can negatively interfere with performance on new tasks, especially when tasks are contradictory.

BFSF combines two complementary policies: a "fast" policy that learns quickly from recent data to mitigate interference, and a "slow" policy that uses meta-reinforcement learning to capture knowledge from all previous tasks for cross-task generalization. A Bayesian estimation mechanism dynamically selects between these policies based on their expected performance using recent return history.

The framework also incorporates a dual-reset mechanism to prevent performance degradation in neural networks and a data relabeling technique to enhance learning efficiency by augmenting recent data with relabeled historical experiences. Experimental results on MuJoCo and Meta-World environments demonstrate that BFSF effectively mitigates interference while maintaining good cross-task generalization, outperforming existing baseline methods in non-stationary settings.

**Questions:**

1) How does the computational overhead of maintaining dual policies, meta-RL training, and Bayesian inference scale with the number of tasks and complexity of environments?

2) How sensitive is the framework to key hyperparameters like the window size (w=100), reset frequency (ν=50), and relabeling percentage (50%)?

3) Since the method assumes unknown task boundaries, how would it perform in scenarios with gradual task transitions or when task changes are very subtle?

4) Would BFSF be effective in other domains like discrete action spaces, vision-based tasks, or environments with fundamentally different dynamics than the tested locomotion tasks?

**Ethical Concerns:**

["NO or VERY MINOR ethics concerns only"]

**Limitations:**

yes

**Paper Formatting Concerns:**

The format of the paper seems in accordance with the format stated in the website.

**Quality:**

3

**Strengths And Weaknesses:**

Strengths:
1) Addresses the understudied but critical issue of cross-task interference in non-stationary MDPs
2) The dual-policy approach elegantly balances fast adaptation (recent data) with cross-task generalization (historical data), with principled Bayesian selection between them.
3) Thorough experimental validation across multiple environments

Weaknesses:
1) Experiments are restricted to a small set of MuJoCo and Meta-World environments with fixed task switching intervals
2) The framework requires maintaining dual networks, meta-RL training, Bayesian inference, and data relabeling, which likely increases computational complexity compared to simpler baselines.
3) Relies on assumptions like normal distribution for Bayesian inference and requires a learned dynamics model for data relabeling

---

> ### Author Rebuttal · Authors · 2025-07-31
>
> Thanks for the valuable review.
>
> **Q1** How does the computational overhead of maintaining dual policies, meta-RL training, and Bayesian inference scale with the number of tasks and complexity of environments?
>
> **A1** Thank you for the question. The computational overhead of our method remains modest and does not scale with the number of tasks, since we always maintain only two policies (fast and slow), regardless of how many tasks are encountered. We do not allocate a separate policy per task; instead, generalization is achieved through a context-based conditioning mechanism.
>
> The Bayesian inference module only performs selection between the two policies based on recent returns, and thus its overhead is negligible and does not scale with task number or environment complexity.
>
> Compared to context-based baselines (e.g. LILAC), the primary additional cost comes from training and maintaining extra 'fast' dual policies. In practice, this results in only a moderate increase in training time.
>
> **Q2** How sensitive is the framework to key hyperparameters like the window size (w=100), reset frequency (ν=50), and relabeling percentage (50%)?
>
> **A2** Appendix B.3 provides sensitive studies on the window of recent data $w$, the reset frequency $\nu$ and the relabel percentage. Additionally, we present the following study for reference:
>
> **Sensitivity study on the window of recent data $w$:** (For reference, the average return of the baseline (LILAC) is $757.8$.)
>
> |  | $w=10$ | $w=50$ | $w=100$ | $w=150$ | $w=200$ |
> | ---- | ---- | ---- | ---- | ---- | ---- |
> | BFSF | $1163.3$ | $1273.6$ | $1209.0$ | $1121.1$ | $945.5$ |
>
> **Sensitivity study on the reset frequency $\nu$:**
>
> ||$\nu=10$|$\nu=25$|$\nu=50$ | $\nu=75$ | $\nu=100$ |
> |----|---- | ---- | ---- | ---- | ---- |
> |BFSF|$1250.6$ | $1169.0$ | $1209.0$ | $1132.1$ | $1116.6$ |
>
> **Sensitivity Study on the relabel percentage:**
> |Relabel Percentage (%)|0|10|20|30|40|50|60|70|80|90|100|
> |----|----|----|----|----|----|----|----|----|----|----|----|
> |Average Return|1055.5|954.9|1046.3|1097.5|1186.4|1039.4|1217.7|1200.9|1148.1|1027.6|1129.1|
>
> **Q3** Since the method assumes unknown task boundaries, how would it perform in scenarios with gradual task transitions or when task changes are very subtle?
>
> **A3** The experiments in gradual task transitions are provided in Appendix B.3. Additionally, we present the result for reference:
>
> ||BFSF|Baseline: LILAC|
> |----|----|----|
> |Gradual Ant-Goal|$-328.5\pm2.1$|$-619.2\pm3.7$|
>
> **Q4** Would BFSF be effective in other domains like discrete action spaces, vision-based tasks, or environments with fundamentally different dynamics than the tested locomotion tasks?
>
> **A4** BFSF is designed to address the interference problem in non-stationary MDPs, a challenge that may also arise in domains like discrete action spaces and vision-based tasks. We appreciate the valuable suggestion and agree that exploring these domains will be crucial for further validation.

---

> > ### Comment · Reviewer_fHvL · 2025-08-08
> >
> > Thank you for your response and clarifications! I think this is a strong paper, and will continue to recommend acceptance.

---

### Decision · Program_Chairs · 2025-09-17

**Decision:**

Accept (poster)

**Comment:**

This paper proposes a Bayesian Fast-Slow Framework (BFSF) for mitigating interference in non-stationary reinforcement learning. The framework combines a fast policy for rapid adaptation, a slow policy based on PEARL for generalization, and a Bayesian mechanism to select between them. Additional techniques (dual-reset and data relabeling) are introduced to stabilize training and accelerate adaptation. Experiments on MuJoCo and Meta-World benchmarks demonstrate clear improvements over strong baselines, with ablations highlighting the contributions of individual components.

The reviewers appreciated the principled design, novelty of the fast/slow Bayesian integration, and strong empirical results on challenging non-stationary tasks. The rebuttal and discussion clarified initial concerns about missing baselines, limited seeds, and theoretical presentation, with the authors providing additional experiments and improved explanations. While the theoretical component remains relatively weak and the empirical evaluation could be broader, the main ideas are well-motivated and convincingly validated.

Overall, this is a solid contribution to continual and meta-RL, addressing an important problem with a clear and practical solution. The paper is not quite at the level of a spotlight or oral due to its limited theoretical depth and scope, but it makes a valuable and well-executed contribution.